# Development of Leisure Valuation Assessment Tool for the Elderly

**DOI:** 10.3390/ijerph19116678

**Published:** 2022-05-30

**Authors:** Da Sol Park, Hae Yean Park

**Affiliations:** 1Department of Occupational Therapy, Jeonju Kijeon College, Jeonju 54989, Korea; otdasol@gmail.com; 2Department of Occupational Therapy, College of Software Digital Healthcare Convergence, Yonsei University, Wonju 26493, Korea

**Keywords:** assessment tool, leisure activities, model fit, occupational therapy, reliability, the elderly, validity, values

## Abstract

This study aimed to develop a leisure valuation assessment tool to revitalize leisure activities for the elderly living in the community. The research method, literature review, and Delphi survey were conducted for the expert panel. Then, the leisure value and participatory leisure activity items were derived to form the assessment items. The two Delphi surveys revealed 38 leisure value assessment items and 41 participating leisure activity items. We attempted to verify the model suitability and validity of the leisure value assessment items through confirmatory factor analysis. The verification showed a good fit. Based on the intensive validity test result, AVE (average variance extracted) values were 66 for physical leisure activities, 65 for emotional leisure activities, and 65 for social leisure activities. The conceptual reliability was 0.96 for physical leisure activities, 0.95 for emotional leisure activities, and 0.96 for social leisure activities. Regarding the internal consistency for reliability verification, Cronbach’s alpha values for physical leisure, emotional leisure, and social leisure activities were 0.909, 0.925, and 0.955, respectively. Hence, the items were highly interrelated and homogeneous tests that measured the same characteristics. The assessment tool can be used to identify useful information on the leisure activities of the elderly and to activate leisure activities for the elderly.

## 1. Introduction

Leisure refers to non-forced and internally motivated activities that involve free time other than mandatory participation in work, self-management, or sleep [1]. From a life cycle perspective, leisure activities play an important role at all ages; however, they have greater significance for the elderly. Leisure activities in old age are not simply concepts of rest, but activities to improve the quality of life in old age, which positively affects retirement life and helps to slow aging [2]. According to the 2019 National Leisure Activity Survey, leisure activity participation time in the 30s to beyond the 70s tends to increase with age; however, the number of leisure activities participated in is decreasing [3]. This means that the diversity of leisure activities for the elderly is decreasing and not being activated. To present leisure activities that meet the needs of the elderly, it is necessary to grasp the values pursued by them when participating in leisure activities. Value refers to the qualitative factors that the participant thinks are important and correct [4,5]. It is also defined as the fundamental attitude toward the world, including oneself, or the ideas therein [6]. Accordingly, leisure value—a perspective formed based on one’s assessment of the significance and role of leisure in an environment including individuals—can be said to be an important qualitative factor when participating in leisure activities [7].

These leisure values were continuously evaluated in the past; they are clearly distinguishable from quantitative assessments, such as leisure activity participation time, frequency, and degree of performance in existing studies [8,9,10,11]. Studies related to leisure value assessment were conducted about 40 years ago. They include the Elder Version of Leisure-Time Activity Enjoyment Scale Assessment Tool for Leisure Tendency of Elder Adults, Leisure Time Situation Scale, and Leisure Activity Party Scale [9,12,13].

However, the existing assessment tools related to leisure values have limitations. First, it is difficult to use these tools designed for evaluating specific leisure activities and has limited opportunity to present new leisure activities that meet the needs of the elderly. Second, since the participants of the assessment are diverse, such as foreigners, adults, and the disabled, it cannot be said to be an elderly-oriented assessment tool. Hence, there is a limit to its application to the elderly living in domestic communities.

Therefore, this study aimed to (1) develop a “leisure valuation assessment tool for the elderly” (LVAT-E) to revitalize leisure activities for the elderly living in the community and (2) verify the suitability, reliability, and validity of this model’s assessment tool for the elderly living in the community.

## 2. Methods

This study was conducted in two stages. The first step was to collect preliminary items and organize the items of the LVAT-E through Delphi research. The second step was to develop assessment tools through model suitability, reliability, and validity verification (Figure 1). This study was approved by the Yonsei University Future Campus Bioethics Review Committee (1041849-202007-BM-089-03).

### 2.1. Composition of Assessment Items

To construct the assessment items, related prior studies were considered, and Delphi surveys were conducted on a group of experts in related occupations. Based on the derived results, leisure value assessment and participatory leisure activity items were included.

#### 2.1.1. Literature Review and National Leisure Activity Survey

As this study was conducted in Korea, mainly journals that included relatively many Korean studies were searched for the literature review. We searched the databases of Pubmed, Google Scholar, and Riss, and the search terms were “Leisure” and “Assessment” or “Measure” or “Scale”. The detailed criteria for selection and exclusion were as follows.
-Selection criteria
(1)Research published in academic journals in the last 10 years (2011–2020);(2)Study written in Korean or English;(3)A study on the Quality Assessment Tool for Leisure Activities for the Elderly.-Exclusion criteria
(1)A study on the evaluation of specific leisure activities;(2)A paper that is impossible to read in full;(3)Research in the forms of meta-analyses, degree theses, books, and posters.

##### Classification Criteria for Leisure Activities for the Elderly

The same classification was used in this study based on the classification of elderly leisure activities into physical, emotional, and social leisure activities using elderly activity theory, continuous theory, and social exchange theory, which are related to elderly leisure [14,15] (Table 1).

#### 2.1.2. Delphi Survey

A Delphi survey was conducted to construct the assessment items. The Delphi survey consisted of 25 people who had more than seven years of clinical and educational experience with the elderly and community occupational therapy or a master’s degree in occupational therapy. Detailed information on the experts is presented in Table 2. The Delphi survey in this study was conducted twice via e-mail in August 2020. In this study, a modified Delphi technique using a structured questionnaire different from the first Delphi was used [16].

Fitness was evaluated on a 4-point Likert scale (1 point: very inappropriate, 4 points: very appropriate). In the data analysis, the content validity ratio (CVR), average, standard deviation, stability, convergence, and consensus were analyzed for the responded content.

### 2.2. Development of Assessment Tool

#### 2.2.1. Application of Assessment Tools

After the Delphi survey, an assessment tool was applied to 454 elderly (aged ≥ 60 years) living in the community from August to September 2020. The number of participants was calculated based on the findings of Mitchell (1993), which stated that a sample size of at least 10 times the number of observation variables was required. The selection criteria were as follows, and general information is shown in Table 3 [17]:-A community resident aged 60 or older.-A person who has not been diagnosed with dementia, cognitive impairment, etc., and who can understand the contents of the evaluation tool.

In principle, a tool is a self-checklist and involves offline implementation, but it was implemented in the form of an online survey through research companies due to environmental constraints caused by COVID-19. Online explanations and consent forms for the study subjects were presented, and only those who pressed the study consent button participated in the study. SPSS 25 was used for the statistical analysis, and descriptive statistics and one-way analysis of variance were used.

#### 2.2.2. Reliability

##### Verification of Internal Consistency

In general, in the field of social science, the internal consistency is judged as “acceptable,” “good,” and “very good” when it is ≥0.6, ≥0.7, and ≥0.8, respectively [17].

#### 2.2.3. Construct Validity

Confirmatory factor analysis using Analysis of Moment Structures (AMOS) was conducted to verify construct validity. Confirmatory factor analysis is useful for measuring construct validity because it can evaluate the overall fit of the model and measure the factor load between observations and latent variables. The construct validity verification procedure was performed in the following order: model suitability, convergent validity verification, and discriminant validity verification.

#### 2.2.4. Assessment of Utility

The effectiveness of the assessment tool was evaluated by applying it to 13 elderly people living in the community. The criteria for selecting participants were as follows:-A community resident aged 60 or older.-A person who has not been diagnosed with dementia, cognitive impairment, etc., and who can understand the contents of the evaluation tool.

The utility test was evaluated on a 5-point Likert scale for item understanding, assessment method understanding, and appropriateness of writing time.

## 3. Results

### 3.1. Item Composition Result

#### 3.1.1. Literature Review and National Leisure Activity Survey Results

Of the collected items, 39 were about leisure value and 45 about participating in leisure activities. They were derived by integrating similar concepts and deleting overlapping items (Table 4 and Table 5).

#### 3.1.2. Delphi Survey Results

Following the analysis of the response values of two Delphi surveys, all items in parts 1 and 2 showed significant values of the minimum value with a CVR of ≥0.37, convergence of 0.5, agreement of ≥0.75, and stability of ≤0.8 (Table 6). Thirty-eight leisure value assessment items and forty-one leisure activities were assessed.

### 3.2. Results of Developing Assessment Tools

#### 3.2.1. Construct Validity Results

##### Model Fit Results

The model suitability results showed good suitability; however, the GFI was 0.825, which is slightly above the standard of 0.8 (Table 7).

##### Convergence Validity Results

The AVE values and conceptual reliability values are shown in Table 8.

##### Discriminant Validity Results

The discriminant validity analysis showed that the AVE of all the corresponding observation variables was larger than the square of the correlation coefficient (Table 9).

#### 3.2.2. Reliability Results

##### Internal Consistency Results

The internal match analysis revealed that all three areas of the leisure activity sub-items had very high reliability (Table 10).

#### 3.2.3. The Results of the Utility Assessment

Based on the result of the utility assessment, it took 10–15 min per person to apply the assessment tool. The results are shown in Table 11.

## 4. Discussion

To revitalize the leisure activities of the elderly living in the community, this study attempted to develop an LVAT-E that can evaluate various leisure values that relate to the leisure activities of the elderly.

The leisure value assessment items of the developed assessment tool have the advantage of being an indicator of what factors the elderly consider important when participating in leisure activities and being able to closely explain the elderly’s desire for leisure activities. This is partially consistent with the argument in foreign studies that participants’ individual characteristics should be identified because they influence the leisure activities they participate in [18,19,20].

The participatory leisure activity items developed in this study have the advantage of being able to separately present “leisure activities they are currently participating in” and “leisure activities that they are not currently participating in but are willing to participate in the future.” If the leisure value assessment items developed in this study and the participatory leisure activity items are used together, the needs of the elderly can be closely understood. Additionally, the participatory leisure activity items can provide practical help when planning leisure activities. Moreso, the LVAT-E can be seen in previous studies as an assessment tool that clearly supports the opinion that qualitative and quantitative factors should be evaluated together when evaluating leisure activities for the elderly [8,9,10,11,12].

The process used to develop the assessment tool in this study had some limitations. First, when applying the assessment tool, the age group was unevenly distributed. In addition, most of the subjects were highly educated. The participants’ age and educational background are factors that influence their participation in leisure activities, and the frequency and type of participation change accordingly [3]. In future studies, reliability and validity should be verified by considering the age group and educational background of the sample.

Second, the assessment tool was applied in the form of an Internet-based survey due to the influence of COVID-19. Since the participants were aged 60 years or older, in future studies, it will be necessary to conduct offline self-checklists when applying assessment tools in consideration of the characteristics of the elderly.

Third, some previous studies used a measurement method and a method of improving the completeness of the assessment scale establishment of the expert advisory meeting when establishing the assessment tool scale. However, this study used a 5-point Likert scale based on previous studies without an expert advisory meeting. In future studies, expert opinions on leisure activities should be reflected in the process of establishing measurement methods and assessment scales.

Despite these limitations, the LVAT-E is not limited to the assessment of specific leisure activities. Additionally, it is possible to evaluate various values pursued by the subject when participating in leisure activities. Hence, the significance of this study is that it can help the subject to plan new leisure activities or suggest a direction in which the leisure activities they are currently participating in should be improved. In modern society, the time for the elderly to participate in leisure activities is increasing; however, the diversity of the leisure activities they participate in is decreasing. It is expected that the assessment tool developed in this study can be used to identify helpful information on the leisure activities of the elderly and to activate leisure activities.

## 5. Conclusions

This study aimed to develop an LVAT-E and verify the reliability and validity of revitalizing leisure activities for the elderly living in the community. The significance of this study is that it can help the elderly to plan new leisure activities or suggest directions in which the leisure activities they are currently participating in should be improved. The assessment tool developed in this study can be used to identify helpful information on the leisure activities of the elderly and to activate leisure activities.

## Figures and Tables

**Figure 1 ijerph-19-06678-f001:**
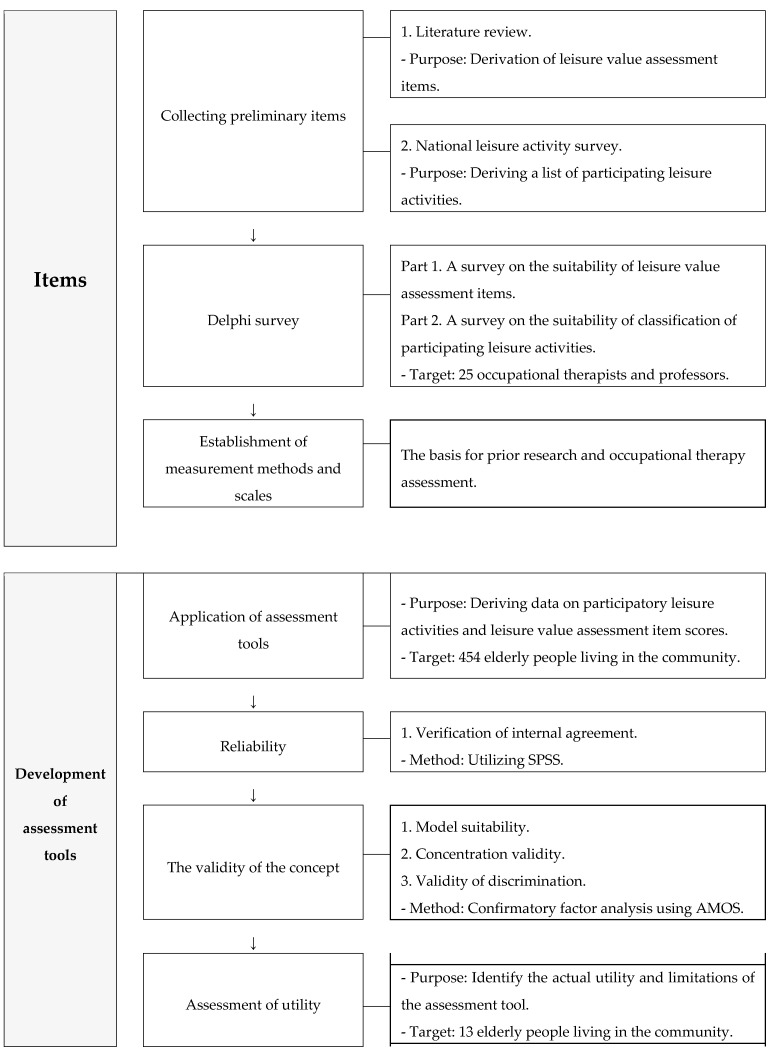
Research process and assessment tool development process.

**Table 1 ijerph-19-06678-t001:** Classification and definition of leisure activities for the elderly.

Classification	Name of Leisure Activities
Physical leisure	Leisure activities that include indoor sports activities and outdoor sports activities and promote physical health.
Emotional leisure	Activities for emotional stability, hobbies for cultural and artistic entertainment, and activities including education through paid and free educational institutions to promote education.
Social leisure	Activities that include community activities or social activities that express roles and beliefs and wills as members of society through them, and activities that return various experiences of life to society.

**Table 2 ijerph-19-06678-t002:** General information of experts.

Career (Month/Year) [Clinical/Education/Research]			Educational Background	Job
3 m 7 y	6 y	6 y	Ph.D.	Professor
6 m 8 y	4 y	-	Ph.D.	Professor
5 y	15 y	-	Ph.D.	Professor
5 y	3 y	4 y	Ph.D.	Professor
4 y	-	6 m	Ph.D.	Occupational therapist
3 y	3 y	-	Ph.D.	Professor
-	-	11 m 4 y	Ph.D.	Researcher
6 m 7 y	-	-	Master’s	Occupational therapist
4 y	2 y	3 y	Ph.D.	Professor
4 y	6 m	-	Ph.D.	Occupational therapist
1 y	-	6 m 3 y	Ph.D.	Professor
1 y	-	6 m 1 y	Master’s	Occupational therapist
3 m 4 y	3 y	4 y	Master’s	Professor
1 y	-	8 m 2 y	Ph.D.	Occupational therapist
6 m	-	2 m 2 y	Master’s	Occupational therapist
1 y	-	8 m 2 y	Ph.D.	Occupational therapist
6 m 5 y	6 m 1 y	8 m 4 y	Ph.D.	Professor
7 y	4 y	-	Ph.D.	Professor
1 y	-	2 y	Master’s	Occupational therapist
4 y	1 y	2 m 2 y	Ph.D.	Professor

**Table 3 ijerph-19-06678-t003:** General information of participants.

Classification		Mean (Standard Deviation)	N	%
Age	60~65	68.12 (3.28)	115	25.3
	66~70		242	53.3
	71~75		75	16.5
	76~80		22	4.8
Final academic background	University		302	66.5
	High school		128	28.2
	Middle school		18	4.0
	Elementary school		5	1.1
	No education		1	0.2
Housemate	Living alone		28	6.2
	Husband and wife		302	66.5
	A married child		20	4.4
	An unmarried child		90	79.8
	Other		14	3.1
Residential area	Big city		34	7.5
	Small- or medium-sized city		420	92.5
Subjective health conditions	Great health		11	2.4
	Good health		120	26.4
	Normal		230	48.5
	Unhealthy		96	21.1
	Very unhealthy		7	1.5

**Table 4 ijerph-19-06678-t004:** The 39 leisure value items found through the literature review [12].

Name of Assessment Tool	Concepts
The Leisure Time Satisfaction Scale (LTS)	- Activities with family - Activities with friends - Social support
Elderly version of Leisure-Time Activity Enjoyment Scale (LAES)	- Achieve an accomplishment - Affirmative change of the mind - Affirmative change of the body - Enjoyment - Pleasure - Socialization
The Leisure Assessment Inventory	- Adaptive behavior - Life satisfaction
Assessment Tool for Leisure Tendency of Older Adults	- Leisure lifestyle - Leisure motivation
Leisure Nostalgia Scale	- Group identity - Leisure experience - Personal identity - Socialization
The Leisure Boredom Scale	- Boredom
Physical Activity and Leisure Motivation Scale	- Affiliation - Appearance - Others’ expectations - Enjoyment - Competition - Ego - Physical condition - Psychological condition
Global Leisure Meanings Scale (GLMS)	- Escaping pressure - Group harmony - Leisure friendship - Passing time - Self-development
Leisure Activity Participation Scale	- Activity with an attractive environment - Developmental - Entertaining - Esthetic - Exciting - Productive - Relaxing - Social

**Table 5 ijerph-19-06678-t005:** The 45 participating in leisure activities items found through the National Leisure Activity Survey.

Classification of Leisure Activities (Physical/Emotional/Social)	Names of Leisure Activities
Physical leisure	- Participation in ball sports - (gateball, soccer, basketball, golf, tennis, badminton, bowling, etc.) - Swimming - Fitness - Aerobics - Stretching - Yoga and pilates - Fishing - Visit zoos, botanical gardens, and amusement parks - Walking and walking - Mountain climbing
Emotional leisure	- Exhibitions, performances, and movies - Watching sports events - Listening to music - Collection activities - Driving a car - Cooking - Taking care of pets - Singing - Taking a photo - Painting - Calligraphy - Playing musical instruments - Gardening - Napping - Watching TV - Listening to the radio - Using Internet media - Writing - Reading discussions - Reading newspapers and magazines - Acquiring language, skills, and certificates - Studying - Going on a trip
Social leisure	- Volunteer activity - Religious activities - Family and relatives - A peer group - Shopping/eating out - Go, chess, hwatu - Chatting, calling, and texting - Games and puzzles - A picnic - Going to a hot spring or a bathhouse - Participate in local festivals and feasts - Going to a senior citizen center

**Table 6 ijerph-19-06678-t006:** Results of the 1st and 2nd Delphi surveys.

	Average	SD	Convergence	Agreements	Stability	CVR
1st Delphi part 1	3.72	0.46	0.48	0.72	0.20	0.78
2nd Delphi part 1	3.43	0.55	0.33	0.82	0.16	0.89
1st Delphi part 2	3.46	0.62	0.42	0.76	0.18	0.80
2nd Delphi part 2	3.55	0.51	0.27	0.86	0.15	0.93

**Table 7 ijerph-19-06678-t007:** Model fit.

χ²	df	CMIN/DF	RMSEA	SRMR	CFI	TLI	GFI
1416.182	479	2.957	0.066	0.0561	0.902	0.892	0.825

**Table 8 ijerph-19-06678-t008:** Convergence validity results.

Sub-Item Classification	AVE	Conceptual Reliability
Physical leisure activity	0.67	0.96
Emotional leisure activity	0.65	0.95
Social leisure activity	0.65	0.96

**Table 9 ijerph-19-06678-t009:** Discriminant validity results.

Sub-Item Classification	Square Correlation Coefficient	AVE
Physical—emotional leisure	0.63	*Φ*^2^ < 66 (physical)
		*Φ*^2^ < 64 (emotional)
Emotional—social leisure	0.57	*Φ*^2^ < 64 (emotional)
		*Φ*^2^ < 65 (social)
Physical—social leisure	0.25	*Φ*^2^ < 66 (physical)
		*Φ*^2^ < 65 (social)

**Table 10 ijerph-19-06678-t010:** Internal consistency results.

Sub-Item Classification	Crohnbach’s Alpha Value
Physical leisure activity	0.909
Emotional leisure activity	0.925
Social leisure activity	0.955

**Table 11 ijerph-19-06678-t011:** Discriminant validity results of utility assessment.

Sub-Item Classification	Very Positive	Positive	Usually	Negative	Very Negative
	*N* (%)	*N* (%)	*N* (%)	*N* (%)	*N* (%)
The level of understanding	6 (46.2)	6 (46.2)	1 (7.6)	0 (0)	0 (0)
Assessment method understanding	7 (53.8)	5 (38.5)	1 (7.6)	0 (0)	0 (0)
Appropriateness of writing time	5 (38.5)	6 (46.2)	1 (7.6)	1 (7.6)	0 (0)

## Data Availability

The data that support the findings of this study are available on request from the corresponding author (H.Y.P.). The data are not publicly available owing to restrictions (e.g., they contain information that could compromise the privacy of the research participants).

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
