# Peer review of "Development of Leisure Valuation Assessment Tool for the Elderly"

_ijerph, 2022, doi:10.3390/ijerph19116678_

Round 1

Reviewer 1 Report

Thank you for the opportunity to review this work. The objective of the study was to develop a leisure valuation assessment tool to revitalize leisure activities for the elderly and to obtain data on its reliability and validity. The objective is interesting and has practical implications for promoting active aging and improving the quality of life of this population group.

However, it would be necessary for the authors to expand the information on some sections and clarify some doubts about the methodology.

LITERATURA REVIEW

  • It is necessary that the authors clarify if they have used any criteria to select the databases and justify why they have not included other commonly used ones (WOS or SCOPUS).
  • It would also be useful to clarify why they introduced "Relaxation Assessment" as a key word and did not include a word focused on the target population group, such as "elderly".
  • Were any inclusion and exclusion criteria considered in the review (e.g., temporal, type of study, population)?

DELPHI SURVEY

  • It would be necessary to have more information on how they selected the group of experts in related occupations who participated in the Delphi surveys, how did they contact them, were there inclusion and exclusion criteria in the selection? If possible, it would also be useful to have some more information on their socio-demographic characteristics (age, gender, place of work).

ITEMS

The authors say: "Of the collected items, 39 were about leisure value and 45 about participating in leisure activities. They were derived by integrating similar concepts and deleting overlapping items." More information on the items and their content is needed.

APPLICATION OF ASSESSMENT TOOL

  • Were there any criteria for the selection of the 454 elderlies to whom the assessment tool was applied? can you include more information about these participants (age, gender, educational level, socioeconomic status, place of residence, among others)?
  •  What objectives were participants told their participation had?
  •  Were measures collected on other variables, such as level of disability, diagnosed illnesses, or emotional problems?

ASSESSMENTE OF UTILITY

  • The authors should indicate whether any criteria were used to select the 13 participants for the assessment of utility.

RESULTS OF DEVELOPING ASSESSMENT TOOLS

  • The authors should define the constructs of physical leisure activities, emotional leisure activities and social leisure activities. For example, driving a car is an emotional activity, why?

In summary, the study is interesting, but there is a lack of information to be able to determine the relevance of the instrument and to be able to verify what this instrument contributes compared to others that already exist (participants and selection criteria and instrument).

Author Response

Thank you very much for taking the time to review my research. We revised our paper by accepting all your opinions. Additionally, we have also included modifications to other judges' comments. The contents are attached in PDF.

Reviewer 2 Report

Overall, a fair paper with some interesting concepts. It is great to see that the authors place value on leisure activities for the elderly and are working toward making this more meaningful. Based on your title, I was expecting to see the assessment tool and have more discussion around its use and benefits in the community. The introduction was well written and highlighted some current challenges for the elderly. The methods and results sections need significant changes - more detail, better linking of information and clearer presentation. Information in the discussion should be in methods/results. The discussion could benefit from more applied thinking about what this study means for elderly people, the community, their families etc... Some specific comments below.  

Line 39/53: I would avoid the term "subject" and say participant or person if you are not discussing research activities. 

Line 75: Figure 1 belongs under the methods introduction section (Line 65) not under the literature review. 

Line 86: Why did you select 454 participants? How was the tool applied? How were participants approached? And what was the tool - can you include the items in the methods section to make it clear what you are testing. 

Line 88: This sentence implies that only 484 elderly people live in the community in question - is this correct? Or is this a representative sample? It would be beneficial to describe the context in the methods section (Setting). 

Line 92-104: I assume that you are testing the Assessment tool from the section above? This needs to be more clear. 

Line 103: What is the utility test?  What assessment tool are you talking about here? 

Line 109: Figure 2 belongs in the methods. It is not results. 

Line 113: Is this all of the information you obtained from the literature review? It would be beneficial to expand on this regarding the assessment items that were selected.  

Results: Would the demographics of the test population change the outcomes? What information was actually collected about the population? 

Lines 125-150: These results are well presented by there is no linking information. It is difficult to follow what you are measuring and why. It would be beneficial to clarify this in the methods (Line 100). 

Lines 172-174 are results (demographics) and should be in the results section. 

Lines 178 - 180: This is your methods and should be in the methods section as per my comment above. 

Author Response

(The authors gave the same response as above.)

Round 2

Reviewer 1 Report

I think the authors have included modifications that have improved the article. In my opinion, they have also incorporated information that clarifies some doubts in the methodology and results section.  The authors have endeavoured to incorporate the reviewers' suggestions or have explained the reasons why they have not. 

However, the authors should check small formatting errors in the text that they have included in this second version (lower case instead of capital letters in some words).

They should also explain what they mean by "an impossible study" when they list the exclusion criteria.

Author Response

According to your comment, we read the full text to make sure there are no errors, and corrected the case-sensitive errors. (ex. Table 4. Capital letters)

In addition, the spelling of the exclusion criteria has been modified, and the explanation of 'an impossible study' have been added.

Exclusion criteria
1) A Study on the Evaluation of Specific Leisure Activities
2) A paper that is impossible to read in full

Reviewer 2 Report

Thank you for addressing my comments in the previous review. Overall, the paper has much improved, well done. Some further minor changes will help the readability. 

Line 53, 142, 198, 213, 218 and 252 - please use "participants" not "subjects". 

Line 75 - I would say "As this study was conducted in Korea" not "since..". This sentence would benefit from a re-word as it is not very clear. 

Please remove "the" from line 114. Otherwise it makes it sound as though there are only 454 elderly people living in the while population. 

Some spacing between paragraphs needs to be fixed. Some headings are in a different font and will need to be fixed up.  

Table 4 - I would use the heading "Concepts" rather than derived items. 

Author Response

I am not fluent in English, so I think there were many technical errors. Thank you very much for your kind comment.

We revised all the parts you said and fixed the spacing between the paragraphs. Also, I modified the font of the table that was different(Ex. Table 1, 4 -).
